# Prediction Model for POstoperative atriaL fibrillAtion in caRdIac Surgery: The POLARIS Score

**DOI:** 10.3390/jcm14020650

**Published:** 2025-01-20

**Authors:** Fabrizio Rosati, Massimo Baudo, Cesare Tomasi, Giacomo Scotti, Sergio Pirola, Giorgio Mastroiacovo, Gianluca Polvani, Gianluigi Bisleri, Stefano Benussi, Lorenzo Di Bacco, Claudio Muneretto

**Affiliations:** 1Division of Cardiac Surgery, Spedali Civili di Brescia, University of Brescia, 25123 Brescia, Italy; rosati.fabri@gmail.com (F.R.); massimo.baudo@icloud.com (M.B.); cesare.tomasi@live.com (C.T.); giacomoscotti16@gmail.com (G.S.); stefano.benussi@unibs.it (S.B.); lorenzo.dibacco@hotmail.it (L.D.B.); 2Department of Cardiovascular Surgery, Centro Cardiologico Monzino, 20138 Milan, Italy; sergio.pirola@ccfm.it (S.P.); gio.mastroiacovo@hotmail.it (G.M.); gianluca.polvani@ccfm.it (G.P.); 3Division of Cardiac Surgery, St Michael’s Hospital, University of Toronto, Toronto, ON M5S 1A1, Canada; gianluigi.bisleri@utoronto.ca

**Keywords:** atrial fibrillation, predictive model, cardiac surgery, postoperative care

## Abstract

**Background**: New-onset postoperative atrial fibrillation (POAF) is the most common complication after cardiac surgery, occurring approximately in one-third of the patients. This study considered all-comer patients who underwent cardiac surgery to build a predictive model for POAF. **Methods**: A total of 3467 (Center 1) consecutive patients were used as a derivation cohort to build the model. The POLARIS score was then derived proportionally from the odds ratios obtained following multivariable logistic regression (MLR). The Brier Score, the area under the receiver operating characteristic curve, and the Hosmer–Lemeshow goodness-of-fit test were used to validate the model. Then, 2272 (Center 2) consecutive patients were used as an external validation cohort. **Results**: In the overall population (n = 5739), POAF occurred in 32.7% of patients. MLR performed in the derivation cohort showed that age, obesity, chronic renal failure, pulmonary hypertension, minimally invasive surgery, and aortic and mitral valve surgery were predictors of POAF. The derived POLARIS score was used to further stratify the population into four risk clusters: low (1.5–3), intermediate (3.5–5), high (5.5–7), and very high (7.5–9), each progressively showing an increase in POAF incidence. This was confirmed in a correlation analysis (Spearman’s rho: 0.636). **Conclusions**: The POLARIS score is a simple-to-use tool to stratify patients at higher risk of POAF. Precise identification of such patients might be used to implement clinical practice with the introduction of preoperative antiarrhythmic prophylaxis, further reducing the incidence of POAF and, potentially, its clinical sequelae, despite further investigations being warranted to test this model in prospective studies.

## 1. Introduction

New-onset postoperative atrial fibrillation (POAF) is the most common complication after cardiac surgery, occurring in approximately one-third of patients, but it can peak at 40–50% after valve-related surgery. POAF is variably associated with increased in-hospital morbidity and mortality, as well as prolonged length of hospital stay and excess hospitalization costs [1,2,3,4]. Atrial structural alterations, pericardial effusion and inflammation, peri-atrial adipose tissue metabolic activity, autonomic neuromodulation, and re-entry and ectopic activity in the pulmonary veins area were identified among the main pathophysiological mechanisms underlying POAF [4]. Efforts have been made to build risk models identifying specific groups of patients at high risk of developing POAF [5,6,7]. However, most of these attempts have been performed in a selected population, thus excluding, for example, patients undergoing complex cardiac procedures. Previous risk models for POAF in cardiac surgery were often based on small cohorts or focused exclusively on CABG patients, excluding high-risk valve surgery populations. Even larger cohort studies incorporated postoperative variables, limiting the ability to predict POAF preoperatively and apply prophylactic treatments, particularly in urgent or emergent cases. Therefore, the present study considered all-comer patients who underwent cardiac surgery in order to determine risk factors implied in the development of POAF. We then set up an additive risk calculator for POstoperative atriaL fibrillAtion in caRdIac Surgery (POLARIS Score) according to the derived odds ratios and performed an external validation to test our hypothesis.

## 2. Materials and Methods

### 2.1. Study Population

Data from adult patients undergoing cardiac surgery with no history of preoperative atrial tachyarrhythmia were retrospectively retrieved from institutional databases at two different European cardiac centers (ASST Spedali Civili di Brescia, University of Brescia, Italy [Center 1] and Centro Cardiologico Monzino IRCCS, Milano, Italy [Center 2]). We did include all patients who underwent elective, urgent, and emergent procedures, as well as complex procedures (more than 2 or 3 procedures/patient). All data, including demographics and clinical and surgical details, have been prospectively recorded and collected at the time of the hospitalization.

The study protocol was approved by the Ethics Committee at Spedali Civili di Brescia, Brescia, Italy, with number NP4980 (15 October 2021) and waived for patient informed consent. This study was reported according to the transparent reporting of a multivariable prediction model for individual prognosis or diagnosis (TRIPOD) guidelines [8] (Appendix A). The data that support the findings of this study are available from the corresponding author upon reasonable request.

### 2.2. Endpoint and Definitions

The primary endpoints of the current study were the incidence and the definition of independent predictors for postoperative AF through regression analysis. The secondary endpoint was to build and test a predictive model of POAF in patients undergoing cardiac surgery.

POAF was defined by the documentation of AF of any duration at any point in the perioperative period before discharge on a rhythm strip or 12-lead electrocardiogram. Body mass index (BMI) was calculated using the formula BMI = weight (kg)/height^2^ (m^2^). Obesity was considered above a BMI of 30 kg/m^2^. The estimated glomerular filtration rate (eGFR) was calculated with the use of the Chronic Kidney Disease Epidemiology Collaboration equation.

### 2.3. Atrial Fibrillation Pharmacological Management

Patients considered in this study were in sinus rhythm without a history of AF and were not in class I or III antiarrhythmic drug (AAD) regimens, while 45.0% (2581/5739) of patients were in class II (b-blockers) or IV (Ca-antagonists) preoperative antiarrhythmic therapy. This regimen was continued until the day of surgery and then routinely restarted at the same, lower, or increased doses according to the patients’ hemodynamics. During the whole hospitalization period, patients were monitored daily until discharge with ECG telemetry and standard 12-lead ECG in case of clinical suspicion of AF. Unless contraindicated, intravenous/oral amiodarone was the first line of therapy in patients experiencing POAF. In case AAD therapy failed to restore sinus rhythm, patients were discharged with oral amiodarone. The anticoagulation regimen was achieved with warfarin, aiming to target an INR between 2 and 3, and continued for at least 1 month. In case of hemodynamic instability due to the POAF, patients underwent electrical cardioversion. Following 30 days from discharge, patients underwent clinical follow-up, and in case of documentation of sinus rhythm restoration, amiodarone and warfarin were discontinued unless indicated by the index surgical procedure.

### 2.4. Statistical Analysis

Clinical data were prospectively recorded on Microsoft Excel (Microsoft Corp., Redmond, WA, USA). Statistical analysis was performed using R (R Project for Statistical Computing, Wien, Austria) version 4.2.1, using the “PredictABEL” package version 1.2-4.

Categorical variables were presented as frequency counts and percentages and compared between groups using the Chi-squared test or Fisher exact test, accordingly. After checking the normality of continuous variables with the Kolmogorov–Smirnov test, they were presented as mean and standard deviation if normally distributed and were compared between groups using the Student’s *t*-test. If they were not normally distributed, median and interquartile ranges were presented and compared between groups using the Mann–Whitney U test. All tests were two-sided, and the alpha level was set at 0.05 for statistical significance.

For the model building, data from Center 1 were used as a derivation dataset, while data from Center 2 were used as external validation. Univariable logistic regression was used for initial variable selection. Only variables with a *p*-value < 0.1, with clinical significance, and that avoided multicollinearity (e.g., creatinine, eGFR, and CKD) were included in the regression models to avoid overfitting. Multivariable analysis was performed and further refined through stepwise logistic regression with forward and backward selection. Akaike’s Information Criterion (AIC) was used for model comparison. The effect size on variables with POAF was estimated by calculating the odds ratio (OR) and 95% confidence interval (CI). The additive risk score was built following a weighted analysis derived from ORs obtained from the multivariable logistic regression from Center 1. Thereafter, the risk factors were ranked from the “strongest” to the “weakest”, and a value ranging from 2 to 0.5 was assigned to each risk factor proportionally in order to obtain the POLARIS score (Appendix A). Then, this score was applied and calculated into the validation dataset (Center 2) for each patient. The Brier Score was calculated to measure the accuracy of probabilistic predictions. Model discrimination was evaluated by using the area under the receiver operating characteristic (ROC) curve. The model was calibrated using the Hosmer–Lemeshow goodness-of-fit test.

Afterward, the predictive ability was calculated as percentage proportions between the number of POAF patients and the total number of patients in each POLARIS score group separately for the two centers; the correlation between the two series of percentages was then calculated using Spearman’s rho coefficient.

## 3. Results

A total of 5739 patients were included in the present study: 3467 from Center 1 and 2272 from Center 2. The flowchart for patient selection can be seen in Figure 1. Patients’ baseline characteristics are depicted in Table 1, while Table 2 and Table 3 summarize operative details and postoperative complications. In the overall population, the median age was 66 years (IQR: 56–73.40), and men were 70.7% (4055/5739). Postoperative AF occurred in 32.7% of patients (1874/5739). Patients suffering from postoperative AF were further analyzed and compared to those who did not experience such complications. The analysis of the two subpopulations showed that postoperative AF was associated with more comorbid patients, as reported in Table 4. They were older (*p* < 0.001) and suffered more from systemic (*p* < 0.001) and pulmonary arterial hypertension (*p* < 0.001), chronic kidney disease (*p* < 0.001), dyslipidemia (*p* = 0.011), and peripheral artery disease (*p* = 0.002).

Intraoperatively, postoperative AF patients underwent more mitral (*p* < 0.001), tricuspid (*p* = 0.011), and aortic (*p* < 0.001) valve surgery and fewer minimally invasive procedures (*p* < 0.001). Postoperatively, besides higher AF episodes, these patients suffered more from blood transfusions (*p* < 0.001), perioperative CVA (*p* < 0.001), need of tracheostomy (*p* = 0.019), severe GI complications (*p* < 0.001), and septicemia (*p* = 0.001). However, 30-day mortality did not significantly differ between the two groups (*p* = 0.227).

The type of cardioplegia influenced POAF rates. Cold cardioplegias showed higher rates of POAF compared to off-pump surgery and warm cardioplegia (Appendix A). At regression analysis these results were confirmed (Appendix A).

### Derivation and Validation Cohorts

For the derivation cohort, significant variables (*p* < 0.1) from the univariable regression were selected for the multivariable analysis. From the multivariable regression, significant variables were further analyzed through stepwise regression (Table 5). The AIC of the multivariable regression and after the stepwise regression were 3827 and 3820.1, respectively. The final OR with 95%CI is depicted in Table 5. For the predictors that were considered in the final model, the Brier score was 0.1833. Appendix A show the population distribution based on the presence of POAF and the predictiveness curve. Discrimination analysis showed an AUC of the ROC of 0.655 (95%CI: 0.634–0.675)—Appendix A. The Hosmer–Lemeshow goodness-of-fit test for model calibration yielded a *p*-value of 0.878 (Figure A1). Appendix A shows the points of the POLARIS score based on the final ORs for each variable. Of note, age is also a well-known risk factor for stroke in AF patients [9], and 65 years of age was taken as the threshold for age grouping [10].

Table 6 depicts patients’ distribution in the derivation, validation, and overall groups for each POLARIS score value and the prevalence of POAF stratified accordingly, Central Picture. A correlation analysis was conducted between the two centers to evaluate POAF percent by POLARIS score. The Spearman’s rho was 0.636. In the validation cohort, the Brier score was 0.2348, the AUC of the ROC was 0.635 (95%CI: 0.61–0.66), and the Hosmer–Lemeshow goodness-of-fit test *p*-value was 0.5543 (Appendix A and Figure A2, respectively). The predictiveness curve is shown in Appendix A.

A correlation between the present predictive score and the CHA_2_DS_2_-VASc score was performed to strengthen the validity of POLARIS. The regression analysis reported an estimate of 0.248 ± 0.013 (*p* < 0.001), thus suggesting a positive correlation between the two.

## 4. Discussion

Atrial fibrillation remains the most frequent complication in patients undergoing cardiac surgery, increasing the incidence of perioperative morbidity and mortality and reflecting a prolonged length of stay and increase in hospitalization costs. Different risk factors have been identified, and efforts were mainly directed towards the prediction and prevention of AF in surgical patients [4]. Pharmacological prophylaxes showed promising results, although, in non-selected patients, the incidence of adverse events could overcome the potential benefits, exposing low-risk patients to unnecessary toxicity and costs [11,12,13,14]. Despite the safety and effectiveness of these strategies, they are still a matter of debate. A reliable bedside tool of risk prediction could help stratify cardiac surgery patients at low to high risk of POAF and tailor the use of preventive pharmacological strategies (Figure 2).

Previously reported risk models for POAF in cardiac surgery patients had a small cohort and restricted their analysis only to CABG patients, as shown by Amar et al. [15], thus excluding a large population of valve surgery patients generally recognized as patients at high risk of POAF [5,16,17,18]. In other studies with larger cohorts, Mathew et al. [19] and Magee et al. [18] still limited their investigations to CABG patients, but they also included postoperative variables, thus hampering the possibility of reliably predicting POAF preoperatively and possibly using prophylactic treatments, especially in the case of urgent or emergent operations. Similar to our analysis, Mariscalco et al. [1] used a large cohort of patients undergoing multiple different cardiac surgery operations to build a POAF score that showed moderate discrimination in a validation cohort. Their predictive score was derived from 17,262 patients, had an AUC of 0.64, and had also been externally validated by splitting the overall population into thirds (2/3 for derivation and 1/3 for validation) [20,21]. Other studies have used smaller cohorts of patients and validated their models with bootstrap samples. Tran et al. used a cohort of 999 patients to build a POAF score that was validated in 5000 bootstrap samples with a c-statistic of 0.69 [22]. Mahoney et al. used 624 patients to build a model with a c-statistic of 0.65 that was validated with 1500 bootstrap samples [16].

In the present study, we retrospectively investigated a large population of more than 5700 all-comer patients who underwent cardiac surgery in two different institutions (Center 1, Spedali Civili di Brescia, University of Brescia, Italy—3467 pts; Center 2, Centro Cardiologico Monzino, University of Milan, Italy—2272 pts). All types of interventions, isolated or combined multiple procedures, as well as urgent and emergent cases, were included. Patients with a history of preoperative AF were excluded regardless of the presence of sinus rhythm at the time of hospitalization in order to avoid overestimating by counting and misinterpreting recurrences of a preexisting arrhythmia as a new onset POAF. We intentionally extended our investigation to all comers without preoperative AF who underwent cardiac surgery to reflect a real-world analysis. Moreover, it should be noted that patients with a history of AF undergoing cardiac surgery would most benefit from concomitant ablation procedures [23,24].

In the present analysis, POAF occurred in one-third of the overall population (32.7%), consistent with previous studies where POAF was reported variably, ranging from 20% to 40% in cardiac surgery patients [3].

Multivariable analysis and stepwise logistic regression performed in the derivation cohort (Center 1) highlighted seven preoperative factors influencing the incidence of POAF (Table 5): Mitral valve surgery (either repair or replacement) was the strongest-ranked predictor (OR: 1.90), followed by aortic valve surgery (OR: 1.44), pulmonary hypertension (OR: 1.39), chronic renal failure (OR: 1.31), obesity (OR: 1.20), and age (OR: 1.03), while a minimally invasive approach (either for aortic valve, mitral valve or ascending aorta) was found to be a protective predictor (OR: 0.65). Although age, chronic kidney disease, obesity, and pulmonary hypertension were previously reported as determinants of postoperative AF in cardiac surgery patients, valve surgery, in particular mitral valve surgery, provided per se the highest risk of POAF in our derivation cohort, as similarly reported in a recent review [4]. A diffused atrium injury due to the large atriotomy required during mitral valve surgery has been largely investigated as a sufficient cause of diffuse inflammation able to induce changes in atrial electrophysiology, thus promoting POAF in animal models [3].

Following our derivation analysis, we interestingly found that the use of minimally invasive approaches reduced the incidence of POAF, although this finding remains controversial [25]. While POAF reduction has been reported in some studies [26,27], no difference between minimally invasive and full sternotomy approaches was found by others [28,29]. Both systemic and local inflammation play a role in increasing the incidence of POAF. While the use of the CPB in cardiac surgery was associated with systemic hyper-inflammation and oxidative stress, thus increasing the incidence of POAF, the limited pericardial opening during minimally invasive approaches may locally reduce the release of pro-inflammatory molecules. Although not fully elucidated, the contact between inflammatory cells and cardiac tissue is of paramount importance in the development of POAF in cardiac surgery patients [30,31]. Moreover, in our cohort, a minimally invasive approach was associated with peripheral cannulation for the use of the CPB: avoiding purse-string at the level of the right atrium may further reduce the atrial injury and the subsequent local apoptosis and inflammation induced by the surgical trauma [32,33]. However, a definitive role of peripheral cannulation in the incidence of POAF should require further investigations.

After the identification of the above-mentioned predictors, the following multivariable logistic regression was performed in the derivation cohort (Center 1), and the POLARIS score was then built proportionally with the relative ORs values (Appendix A). We therefore tested the POLARIS in both the derivation and the validation cohorts, obtaining a reliable calibration and discrimination power (Hosmer–Lemeshow *p*-value = 0.878 and 0.554, respectively). A calibration plot visually assesses the agreement between predicted probabilities and observed outcomes by comparing observed event rates to predicted probabilities across risk groups. Ideally, points align closely with the diagonal line, indicating perfect calibration. The Hosmer–Lemeshow test provides a statistical measure of calibration by dividing predictions into groups (generally 10) and comparing observed and expected event rates within each group. A high *p*-value suggests good calibration, while a low *p*-value indicates discrepancies. However, the test’s sensitivity to group size and sample size may limit its reliability, making calibration plots a valuable complementary tool for evaluating model performance.

After stratifying the POLARIS score in 4 clusters of risk (low = 1.5–3; intermediate = 3.5–5; high = 5.5–7; very high = 7.5–9), we demonstrated an increase in the incidence of POAF according to the progression of the categories towards higher risks in the validation cohort. Furthermore, this result was confirmed by means of Spearman’s analysis, identifying a linear positive correlation between the POLARIS score and the rate of POAF in the validation cohort (Spearman’s rho: 0.636). These results might be even more remarkable considering the big differences in terms of preoperative characteristics between the derivation (Center 1) and the validation (Center 2) cohorts, thus suggesting the possibility of widely applying this score to larger, random cardiac surgery populations.

Among other POAF predicting scores, the CHA_2_DS_2_-VASc score was also found to predict POAF [34], even though it was originally designed and currently used to assess the one-year risk of thromboembolic events in non-anticoagulated individuals with atrial fibrillation [35]. Patients at higher risk (CHA_2_DS_2_-VASc > 3) are suggested to receive a prophylactic treatment [20,21,34]. As a result, a comparison between the POLARIS and the CHA_2_DS_2_-VASc score was performed. The regression analysis revealed a significant correlation between the two scores, thus empowering the validity of the POLARIS.

Finally, we investigated the impact of the type of cardioplegia on POAF rates. We found that cold cardioplegic solutions have a higher incidence of POAF compared to off-pump or warm cardioplegia, which is in line with a previous study [36]. Therefore, according to these results, warm cardioplegia might be preferred in patients at high risk of POAF. However, this factor is difficult to mitigate, being the type of myocardial protection a “surgeon preference” and difficult to modify.

### 4.1. Future Perspectives

Further research will be started in our institution by combining AAD prophylaxis in high-risk patients in order to investigate whether or not this therapy may impact the incidence of POAF in this subset of patients. Indeed, current literature recommends the use of AAD to reduce the incidence of POAF in patients considered at high risk [4,37]; however, this recommendation is infrequently applied in practice. The POLARIS score aims to help identify higher-risk patients, potentially encouraging physicians to implement such prophylactic measures.

In high-risk patients, the lack of AF symptomatology should not preclude the use of specific precautions. Indeed, it is fundamental to emphasize that POAF holds significant clinical relevance even in the absence of AF-related symptoms, as asymptomatic patients face the same adverse outcomes as those who are symptomatic [38].

Lastly, considering the positive results obtained in the current study, we would encourage further investigation and confirmation of whether the use of a minimally invasive approach in high-risk patients could be beneficial.

### 4.2. Limitations

The POLARIS score model showed good accuracy when tested on the validation cohort. Moreover, this model was mainly based on and developed on preoperative clinical factors, but the multifactorial etiology of POAF may require implementation with preoperative AF-specific echocardiographic (such as atrial dimensions or diastolic dysfunction), biomarkers (like BNP, neutrophil–lymphocyte ratio, or genetic predisposition to excessive fibrosis), and hemodynamic data that, given the retrospective nature of the study, were not available. Regarding echocardiographic data, it is important to note that ejection fraction was tested in the univariable regression analysis but was found to be non-significant. Valve insufficiencies were not included in the model because they were closely related to the type of procedure, and including them would have introduced multicollinearity. Future prospective studies may help to further improve the predictivity of the score. Additionally, the model needs to be tested prospectively in order to enhance its clinical and fine-tuning usefulness for follow-up AF monitoring.

## 5. Conclusions

The POLARIS score is a simple-to-use bedside tool and may represent a user-friendly predictive score for POAF. It showed a good ability to stratify the risk of POAF in all-comer patients who underwent cardiac surgery. Precise identification of patients at high and very high risk of developing AF postoperatively may implement clinical practice with the introduction of preoperative antiarrhythmic prophylaxis, further reducing the incidence of POAF and, potentially, its clinical sequelae. This would be more relevant, particularly in patients with diastolic function impairment in whom atrial contribution during diastole is of paramount importance to maintain an adequate cardiac output during the postoperative period.

## Figures and Tables

**Figure 1 jcm-14-00650-f001:**
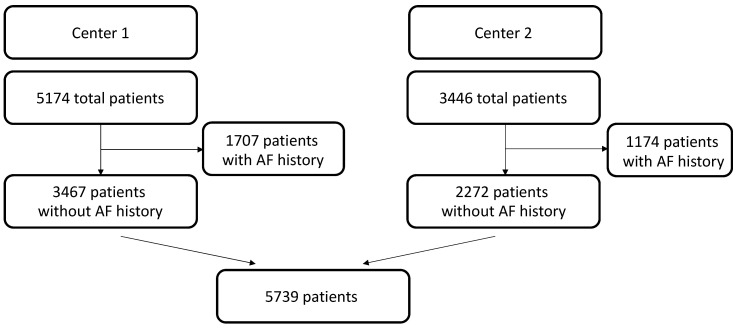
Patient selection flowchart. Institutional databases at two different European cardiac centers (ASST Spedali Civili di Brescia, University of Brescia, Italy [Center 1] (n = 5174) and Centro Cardiologico Monzino IRCCS, Milano, Italy [Center 2] (n = 3446) were retrieved. Patient selection excluded history of AF. AF = atrial fibrillation.

**Figure 2 jcm-14-00650-f002:**
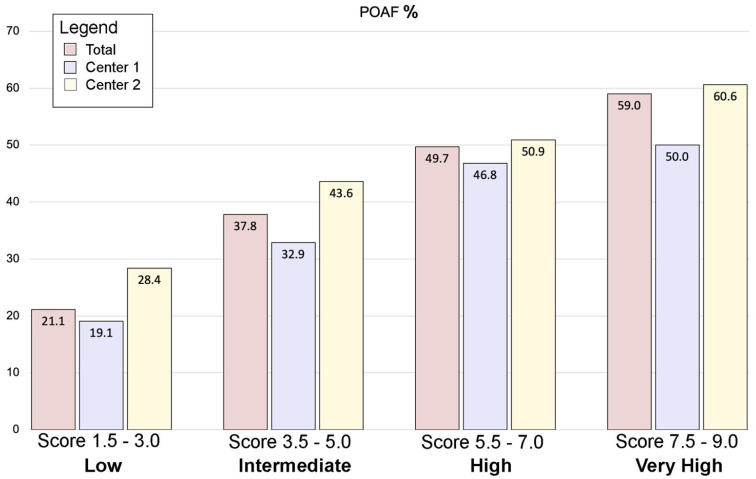
The POLARIS score. A total of 5739 patients were included in the present study: 3467 from Center 1 and 2272 from Center 2. POAF occurred in 32.7% of patients (1874/5739). After model optimization, seven variables were identified to significantly influence POAF and were used to construct the POLARIS score. The four clusters of increasing risk showed a progressive increase in POAF incidence. POAF = postoperative atrial fibrillation.

**Table 1 jcm-14-00650-t001:** Patients’ baseline demographics.

Variable	Overall (n = 5739)	Center 1 (n = 3467)	Center 2 (n = 2272)	*p*-Value
Age	66 [56–73.40]	65.40 [54.00–73.20]	67.00 [58.00–74.00]	<0.001
Male sex	4055 (70.7%)	2525 (72.8%)	1530 (67.3%)	<0.001
BMI	25.95 [23.00–28.74]	26.00 [23.03–29.00]	25.00 [23.00–28.00]	0.001
Obesity	1003 (17.5%)	634 (18.3%)	369 (16.2%)	0.051
Creatinine	0.90 [0.80–1.09]	0.90 [0.80, 1.10]	0.93 [0.80–1.07]	0.003
eGFR	81.77 [64.84–93.33]	83.17 [64.41–94.95]	79.88 [65.39–90.90]	<0.001
CKD	597 (10.4%)	315 (9.1%)	282 (12.4%)	<0.001
History of smoke	1861 (32.4%)	888 (25.6%)	973 (42.8%)	<0.001
CAD familiarity	1277 (22.3%)	806 (23.2%)	471 (20.7%)	0.027
CAD	2570 (44.8%)	1734 (50.0%)	836 (36.8%)	<0.001
Previous PTCA	631 (11.0%)	389 (11.2%)	242 (10.7%)	0.528
Diabetes	1089 (19.0%)	701 (20.2%)	388 (17.2%)	0.005
Dyslipidemia	3055 (53.2%)	1794 (51.7%)	1261 (55.5%)	0.006
Hypertension	3980 (69.4%)	2452 (70.7%)	1528 (67.3%)	0.006
AAD	2581 (45.0%)	1249 (36.0%)	1332 (58.6%)	<0.001
Previous CVA	200 (3.5%)	198 (5.7%)	2 (0.1%)	<0.001
PAD	629 (11.0%)	366 (10.6%)	263 (11.6%)	0.244
Endocarditis	170 (3.0%)	105 (3.0%)	65 (2.9%)	0.779
COPD	636 (11.1%)	458 (13.2%)	178 (7.9%)	<0.001
Cardiogenic shock	49 (0.9%)	41 (1.2%)	8 (0.4%)	0.001
NYHA III-IV	1915 (33.4%)	1559 (45.0%)	356 (15.7%)	<0.001
CHA_2_DS_2_-VASc	2.0 [1.0–3.0]	2.0 [1.0–3.0]	2.0 [1.0–3.0]	<0.001
EF (%)	59.00 [52.00–64.00]	55.00 [50.00–60.00]	62.00 [56.00–66.00]	<0.001
PAH	1788 (31.2%)	550 (15.9%)	1238 (54.5%)	<0.001

AAD = antiarrhythmic drugs; BMI = body mass index; CAD = coronary artery disease; CKD = chronic kidney disease; COPD = chronic obstructive pulmonary disease; CVA = cerebrovascular accident; EF = ejection fraction; eGFR = estimated glomerular filtration rate; PAD = peripheral artery disease; PAH = pulmonary arterial hypertension; PTCA = percutaneous transluminal coronary angioplasty.

**Table 2 jcm-14-00650-t002:** Intraoperative outcomes.

Outcome	Overall (n = 5739)	Center 1 (n = 3467)	Center 2 (n = 2272)	*p*-Value
Non-elective	382 (6.7%)	258 (7.4%)	124 (5.5%)	0.004
Prior Cardiac Surgery	451 (7.9%)	232 (6.7%)	219 (9.6%)	<0.001
Minimally invasive Approach	639 (11.1%)	512 (14.8%)	127 (5.6%)	<0.001
CXC time	75.00 [55.00–101.00]	67.00 [47.25–90.00]	84.00 [64.00–114.00]	<0.001
CPB time	104.00 [81.00–137.00]	98.00 [74.00–126.50]	114.00 [89.00–150.00]	<0.001
CABG	2348 (40.9%)	1627 (46.9%)	721 (31.7%)	<0.001
*On-pump*	1795 (76.4%)	1126 (69.2%)	669 (92.8%)	<0.001
*Off-pump*	553 (23.6%)	501 (30.8%)	52 (7.2%)	
Aortic valve surgery	1960 (34.2%)	956 (27.6%)	1004 (44.2%)	<0.001
*Biologic valve*	1498 (76.4%)	573 (59.9%)	925 (92.2%)	<0.001
*Mechanical valve*	462 (23.6%)	383 (40.1%)	79 (7.8)	
Mitral valve surgery	1383 (24.1%)	592 (17.1%)	791 (34.8%)	<0.001
*Valvuloplasty*	964 (69.7%)	438 (74.0%)	526 (66.5%)	0.002
*Biologic valve*	274 (19.8%)	52 (8.8%)	222 (28.1%)	<0.001
*Mechanical valve*	145 (10.5%)	102 (17.2%)	43 (5.4%)	<0.001
Tricuspid valve surgery	240 (4.2%)	67 (1.9%)	173 (7.6%)	<0.001
*Valvuloplasty*	231 (96.3%)	65 (97.0%)	166 (95.9%)	0.698
*Biologic valve*	9 (3.7%)	2 (3.0%)	7 (4.1%)	
Pulmonary valve surgery	40 (0.7%)	37 (1.1%)	3 (0.1%)	<0.001
Aortic surgery	795 (13.9%)	498 (14.4%)	297 (13.1%)	0.166
*Root*	152 (2.6%)	118 (3.4%)	34 (1.5%)	<0.001
*Ascending aorta*	581 (10.1%)	337 (9.7%)	244 (10.7%)	0.227
*Aortic arch*	62 (1.1%)	43 (1.2%)	19 (0.8%)	0.188

CABG = coronary artery bypass grafting; CPB = cardiopulmonary bypass; CXC = cross-clamp.

**Table 3 jcm-14-00650-t003:** Postoperative outcomes.

Outcome	Overall (n = 5739)	Center 1 (n = 3467)	Center 2 (n = 2272)	*p*-Value
IABP	132 (2.3%)	81 (2.3%)	51 (2.2%)	0.892
Patients transfused	2314 (40.3%)	1289 (37.2%)	1025 (45.1%)	<0.001
Revision for bleeding	203 (3.5%)	113 (3.3%)	90 (4.0%)	0.182
Perioperative AMI	40 (0.7%)	9 (0.3%)	31 (1.4%)	<0.001
Sternal revision	95 (1.7%)	53 (1.5)	42 (1.8)	0.410
Septicemia	87 (1.5%)	26 (0.7%)	61 (2.7%)	<0.001
Severe GI complications	151 (2.6%)	79 (2.3%)	72 (3.2%)	0.039
Perioperative CVA	66 (1.2%)	40 (1.2%)	26 (1.1%)	0.974
Pneumonia	60 (1.0%)	23 (0.7%)	37 (1.6%)	0.001
Atrial fibrillation	1874 (32.7%)	920 (26.5%)	954 (42.0%)	<0.001
PM placement	193 (3.4%)	96 (2.8%)	97 (4.3%)	0.002
Cardiac arrest	23 (0.4%)	12 (0.3%)	11 (0.5%)	0.551
Cardiac tamponade	70 (1.2%)	39 (1.1%)	31 (1.4%)	0.419
Tracheostomy	34 (0.6%)	23 (0.7%)	11 (0.5%)	0.491
MOF	28 (0.5%)	8 (0.2%)	20 (0.9%)	0.001
Hospital stay	7.00 [6.00, 9.00]	6.00 (5.00–8.00)	7.00 (7.00–10.00)	<0.001
30-day mortality	78 (1.4%)	32 (0.9%)	46 (2.0%)	0.001

AMI = acute myocardial infarction; CVA = cerebrovascular accident; GI = gastrointestinal; MOF = multi-organ failure; PM = pacemaker.

**Table 4 jcm-14-00650-t004:** Patients’ demographics and perioperative outcomes according to the occurrence of postoperative AF.

Variable	No AF (n = 3865)	AF (n = 1874)	*p*-Value
*Preoperative*			
Age	64.00 (53.00–72.00)	69.20 (61.92–75.90)	<0.001
Male sex	2807 (72.6%)	1248 (66.6%)	<0.001
BMI	25.91 (23.00–28.73)	26.00 (23.00–29.00)	0.655
Obesity	654 (16.9%)	349 (18.6%)	0.120
Creatinine	0.90 (0.80–1.10)	0.90 (0.80–1.08)	0.625
eGFR	83.45 (65.65–94.94)	78.46 (63.01–90.09)	<0.001
CKD	352 (9.1%)	245 (13.1%)	<0.001
History of smoke	1267 (32.8%)	594 (31.7%)	0.428
CAD familiarity	858 (22.2%)	419 (22.4%)	0.919
CAD	1767 (45.7%)	803 (42.8%)	0.043
Previous PTCA	440 (11.4%)	191 (10.2%)	0.191
Diabetes	741 (19.2%)	348 (18.6%)	0.610
Dyslipidemia	2012 (52.1%)	1043 (55.7%)	0.011
Hypertension	2618 (67.7%)	1362 (72.7%)	<0.001
AAD	1628 (42.1%)	953 (50.9%)	<0.001
Previous CVA	149 (3.9%)	51 (2.7%)	0.034
PAD	389 (10.1%)	240 (12.8)	0.002
Endocarditis	129 (3.3%)	41 (2.2%)	0.020
COPD	417 (10.8%)	219 (11.7%)	0.332
Cardiogenic shock	35 (0.9%)	14 (0.7%)	0.646
NYHA 3–4	1282 (33.2%)	633 (33.8%)	0.668
CHA_2_DS_2_-VASc	2.0 [1.0–3.0]	3.0 [1.0–3.0]	<0.001
EF (%)	58.00 (51.00–63.00)	60.00 (53.00–65.00)	<0.001
PAH	1018 (26.3%)	770 (41.1%)	<0.001
*Intraoperative*			
Non-elective	267 (6.9%)	115 (6.1%)	0.297
Reintervention	319 (8.3%)	132 (7.0%)	0.122
Minimally invasive	511 (13.2%)	128 (6.8%)	<0.001
CXC time	73.00 (54.00–100.00)	78.00 (59.00–104.00)	<0.001
CPB time	103.00 (79.00–134.00)	107.00 (85.00–143.00)	<0.001
CABG	1639 (42.4%)	709 (37.8%)	0.001
*On-pump*	1225 (74.7%)	557 (78.6%)	0.047
*Off-pump*	414 (25.3%)	152 (21.4%)	
Aortic valve surgery	1193 (30.9%)	767 (40.9%)	<0.001
*Biologic valve*	879 (73.7%)	620 (80.8%)	<0.001
*Mechanical valve*	314 (26.3%)	147 (19.2%)	
Mitral valve surgery	819 (21.2%)	564 (30.1%)	<0.001
*Valvuloplasty*	603 (73.7%)	365 (64.7%)	<0.001
*Biologic valve*	133 (16.2%)	148 (26.2%)	<0.001
*Mechanical valve*	83 (10.1%)	52 (9.1)	0.573
Tricuspid valve surgery	143 (3.7%)	97 (5.2%)	0.011
*Valvuloplasty*	133 (93.0%)	90 (92.8%)	0.947
*Biologic valve*	10 (7.0%)	7 (7.2%)	
Pulmonary valve surgery	27 (0.7%)	13 (0.7%)	0.999
Aortic surgery	381 (9.9%)	220 (11.7%)	0.033
*Root*	98 (2.5%)	54 (2.9%)	0.498
*Ascending aorta*	366 (9.5%)	215 (11.5%)	0.021
*Aortic arch*	45 (1.2%)	17 (0.9%)	0.455
*Postoperative*			
IABP	84 (2.2%)	48 (2.6%)	0.409
Patients transfused	1413 (36.6%)	900 (48.0%)	<0.001
Revision for bleeding	131 (3.4%)	72 (3.8%)	0.427
Perioperative AMI	24 (0.6%)	16 (0.9%)	0.409
Sternal revision	59 (1.5%)	36 (1.9%)	0.323
Septicemia	43 (1.1%)	44 (2.3%)	0.001
Severe GI complications	51 (1.3%)	50 (2.7%)	<0.001
Perioperative CVA	31 (0.8%)	35 (1.9%)	<0.001
Pneumonia	34 (0.9%)	26 (1.4%)	0.102
PM placement	134 (3.5%)	69 (3.7%)	0.679
Cardiac arrest	11 (0.3%)	12 (0.6%)	0.075
Cardiac tamponade	42 (1.1%)	27 (1.4%)	0.248
Tracheostomy	16 (0.4%)	18 (1.0%)	0.019
MOF	13 (0.3%)	15 (0.8%)	0.030
Hospital stay	7.00 (6.00–8.00)	7.00 (7.00–10.00)	<0.001
30-day mortality	58 (1.5%)	20 (1.1%)	0.227

AAD = antiarrhythmic drugs; AMI = acute myocardial infarction; BMI = body mass index; CAD = coronary artery disease; CKD = chronic kidney disease; COPD = chronic obstructive pulmonary disease; CPB = cardiopulmonary bypass; CVA = cerebrovascular accident; CXC = cross-clamp; EF = ejection fraction; eGFR = estimated glomerular filtration rate; GI = gastrointestinal; MOF = multi-organ failure; PAD = peripheral artery disease; PAH = pulmonary arterial hypertension; PM = pacemaker; PTCA = percutaneous transluminal coronary angioplasty.

**Table 5 jcm-14-00650-t005:** Multivariable and stepwise regression.

Variable	Estimate ± SE	*p*-Value	Estimate ± SE	*p*-Value	OR (95% CI)
	*Multivariable*		*Stepwise*		
Age	0.029749 ± 0.003442	<0.001	0.030752 ± 0.003301	<0.001	1.0312 (1.0246–1.0379)
Mitral valve surgery	0.635331 ± 0.111565	<0.001	0.643032 ± 0.109971	<0.001	1.9022 (1.5334–2.3598)
Aortic valve surgery	0.366675 ± 0.089709	<0.001	0.364955 ± 0.089214	<0.001	1.4404 (1.2094–1.7157)
Minimally invasive	−0.436048 ± 0.131267	<0.001	−0.432276 ± 0.129828	<0.001	0.6490 (0.5032–0.8371)
PAH	0.325651 ± 0.108381	0.0026	0.332384 ± 0.106235	0.00176	1.3943 (1.1322–1.7170)
CKD	0.272529 ± 0.132413	0.0395	0.267365 ± 0.130788	0.04093	1.3065 (1.0111–1.6883)
Obesity	0.162595 ± 0.103069	0.0946	0.185355 ± 0.100455	0.06502	1.2036 (0.9885–1.4656)
AAD	0.124381 ± 0.084496	0.1410			
Hypertension	0.100003 ± 0.098719	0.3110			
Diabetes	−0.121208 ± 0.102306	0.2361			
COPD	0.027680 ± 0.122956	0.8219			
Male sex	0.000881 ± 0.090084	0.9922			

AAD = antiarrhythmic drugs; CKD = chronic kidney disease; CI = confidence interval; COPD = chronic obstructive pulmonary disease; PAH = pulmonary arterial hypertension; SE = standard error.

**Table 6 jcm-14-00650-t006:** POAF and POLARIS score distribution in the derivation, validation groups, and total population.

Overall								Validation			Derivation		
POLARIS	Patients	POAF	%POAF	CHA_2_DS_2_-VASc	Patients	POAF	% POAF	Patients	POAF	%POAF	Patients	POAF	%POAF
1.5	122	9	7.38	0	601	135	22.5	120	9	7.50	2	0	0.00
2	685	119	17.37	1	1285	341	26.5	561	98	17.47	124	21	16.94
2.5	154	18	11.69	2	1430	447	31.3	139	16	11.51	15	2	13.33
3	1385	350	25.27	3	1294	491	37.9	1008	226	22.42	377	124	32.89
3.5	173	43	24.86	4	771	316	41.0	150	37	24.67	23	6	26.09
4	1341	480	35.79	5	309	127	41.1	781	251	32.14	560	229	40.89
4.5	125	39	31.20	6	41	14	34.1	80	25	31.25	45	14	31.11
5	990	433	43.74	7	8	3	26.7	400	151	37.75	590	282	47.80
5.5	54	14	25.93	8	0	0	0	22	8	36.36	32	6	18.75
6	486	249	51.23	9	0	0	0	152	71	46.71	334	178	53.29
6.5	11	5	45.45	10	0	0	0	1	0	0.00	10	5	50.00
7	174	92	52.87					47	25	53.19	127	67	52.76
7.5	0	0						0	0		0	0	
8	38	22	57.89					6	3	50.00	32	19	59.38
8.5	0	0						0	0		0	0	
9	1	1	100.00					0	0		1	1	100.00

## Data Availability

The data that support the findings of this study are available from the corresponding author upon reasonable request.

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
