# Peer review of "Prediction Model for POstoperative atriaL fibrillAtion in caRdIac Surgery: The POLARIS Score"

_jcm, 2025, doi:10.3390/jcm14020650_

Round 1

Reviewer 1 Report

Comments and Suggestions for Authors

The manuscript explores a crucial clinical topic by presenting the development and validation of the POLARIS score, a predictive model for postoperative atrial fibrillation (POAF) in cardiac surgery patients. The following limitations should be addressed:

The abstract effectively summarizes the study but could benefit from a more explicit mention of the clinical implications of the POLARIS score. Highlight its utility in decision-making for preoperative risk stratification and postoperative management.

In the introduction, the manuscript lacks sufficient context on competing models, particularly their shortcomings that POLARIS aims to address. This comparison would emphasize the novelty of the proposed model.

The validation cohort is a strength; however, details on data preprocessing and consistency checks between derivation and validation cohorts are insufficient. Please elaborate on how discrepancies, if any, were handled.

The selection of predictive variables is well-detailed, but additional rationale for excluding variables such as diabetes or hypertension (non-significant) from the final model would clarify the process.

The results are robust but require more emphasis on the model's performance metrics in practical terms. For instance, what would be the clinical significance of an AUC of ~0.65? How does this compare to the CHA2DS2-VASc score for the same patient cohorts?

The stratification of POAF risk into four clusters is clear, but a table or graph showing the distribution of patients across these risk clusters would improve readability.

Discussion, To strengthen the utility of the POLARIS score emphasize that POAF is of major importance even in the absence of AF-related symptoms, as asymptomatic patients carry the same adverse outcomes as symptomatic individuals. The following meta-analysis published in the European Heart Journal should be cited: DOI: 10.1093/eurheartj/ehae694.

While the discussion is detailed, it would benefit from a critical analysis of the model's limitations. Specifically, the Hosmer–Lemeshow test's p-value is mentioned as a measure of calibration but lacks context. A discussion on calibration-in-the-large or calibration plots would be useful.

Expand on the practical application of the POLARIS score. For example, what interventions would be tailored for "very high risk" patients, and how might these differ from standard care?

The discussion of minimally invasive surgery as a protective factor is interesting but contradictory findings in the literature require a more balanced interpretation.

Address the integration of echocardiographic data or biomarkers into future iterations of the POLARIS score.

Include calibration plots for the derivation and validation cohorts in the main manuscript instead of supplementary materials.

The manuscript is generally well-written but includes minor typographical errors (e.g., missing articles or conjunctions). A thorough proofreading is recommended.

Ethical considerations are adequately addressed. However, authors with disclosed conflicts of interest should clarify how these did not influence the study outcomes.

Author Response

The manuscript explores a crucial clinical topic by presenting the development and validation of the POLARIS score, a predictive model for postoperative atrial fibrillation (POAF) in cardiac surgery patients. The following limitations should be addressed:

A: We would like to thank the reviewer for the positive words and efforts to review our manuscript.

The abstract effectively summarizes the study but could benefit from a more explicit mention of the clinical implications of the POLARIS score. Highlight its utility in decision-making for preoperative risk stratification and postoperative management.

A: Thank you very much for your suggestion. The abstract was modified accordingly to highlight the utility in decision-making.

In the introduction, the manuscript lacks sufficient context on competing models, particularly their shortcomings that POLARIS aims to address. This comparison would emphasize the novelty of the proposed model.

A: Thank you for your precious comment. We agree that the Introduction could be expanded to address competing models. However, we believe this topic has been thoroughly covered in the Discussion section, and an overly detailed Introduction on this point would risk redundancy. Therefore, we have chosen to extend the Introduction paragraph only to a sufficient extent.

The validation cohort is a strength; however, details on data preprocessing and consistency checks between derivation and validation cohorts are insufficient. Please elaborate on how discrepancies, if any, were handled.

A: Thank you for highlighting the importance of the validation cohort. We agree that details on data preprocessing and consistency checks between the derivation and validation cohorts are crucial for transparency and robustness. This data was clearly presented in the manuscript. The external validation cohort was intentionally selected to represent a different population, which allows us to assess the generalizability of the model across varied settings. Discrepancies between the cohorts, such as differences in baseline characteristics, were not artificially adjusted, as they reflect real-world variability. This is a strength of external validation, as it provides insight into how well the model performs in populations distinct from the derivation cohort.

The selection of predictive variables is well-detailed, but additional rationale for excluding variables such as diabetes or hypertension (non-significant) from the final model would clarify the process.

A: Thank you for this important question. In our model, we prioritized statistical significance during variable selection to ensure a parsimonious and robust design, particularly given the sample size constraints. While variables such as diabetes and hypertension are clinically significant in other contexts, they were excluded because they did not demonstrate statistical significance in our dataset.

The results are robust but require more emphasis on the model's performance metrics in practical terms. For instance, what would be the clinical significance of an AUC of ~0.65? How does this compare to the CHA2DS2-VASc score for the same patient cohorts?

A: Thank you for your insightful comment. The AUC of ~0.65 for our model reflects modest discrimination, which is consistent with other described clinical scoring systems described in the Discussion section, which had their own drawback as previously described.

Regarding the CHA2DS2-VASc score, it has been shown in a meta-analysis to achieve AUC values in the range from 0.59–0.71, REF. 33. This demonstrates that an AUC of ~0.65, while modest, aligns with the performance of other tools in literature, but it applies to a population of general cardiac surgery all-comers. Moreover, the CHA2DS2-VASc score was originally built for stroke prediction, not POAF.

The stratification of POAF risk into four clusters is clear, but a table or graph showing the distribution of patients across these risk clusters would improve readability.

A: Thank you for your comment. Figure 2 has the purpose of show as histogram the distribution of patients across the total population, as well as for Center 1 and Center 2. The distribution of these patients is already described in Table 6.

Discussion, To strengthen the utility of the POLARIS score emphasize that POAF is of major importance even in the absence of AF-related symptoms, as asymptomatic patients carry the same adverse outcomes as symptomatic individuals. The following meta-analysis published in the European Heart Journal should be cited: DOI: 10.1093/eurheartj/ehae694.

A: Thank you for your professional suggestion. This important message was added in the discussion section and the reference was cited.

While the discussion is detailed, it would benefit from a critical analysis of the model's limitations. Specifically, the Hosmer–Lemeshow test's p-value is mentioned as a measure of calibration but lacks context. A discussion on calibration-in-the-large or calibration plots would be useful.

A: Thank you for bringing this to our attention. A brief description of Hosmer-Lemenshow and calibration plots was added in the Discussion section.

Expand on the practical application of the POLARIS score. For example, what interventions would be tailored for "very high risk" patients, and how might these differ from standard care?

A: Thank you for this interesting question. In brief, we are unable to address the reviewer’s question due to the lack of relevant data, as this aspect has not been investigated in the current study.

As mentioned in the "Future Perspectives" section, current guidelines recommend the use of antiarrhythmic drugs for POAF prophylaxis; however, this recommendation is infrequently applied in practice. The POLARIS score aims to help identify higher-risk patients, potentially encouraging physicians to implement such prophylactic measures. However, due to the absence of investigation, the specific therapy for this patient subset remains unknown at this time.

The discussion of minimally invasive surgery as a protective factor is interesting but contradictory findings in the literature require a more balanced interpretation.

A: Thank you for your conscientious comment. The sentence was mitigated.

Address the integration of echocardiographic data or biomarkers into future iterations of the POLARIS score.

A: Thank you for your comment. More details were added to the sentence. Regarding echocardiographic data, it is important to note that ejection fraction was tested in the univariable regression analysis but was found to be non-significant. Valve insufficiencies were not included in the model because they were closely related to the type of procedure and including them would have introduced multicollinearity. More AF-specific echocardiographic data would be necessary.

Include calibration plots for the derivation and validation cohorts in the main manuscript instead of supplementary materials.

A: Thank you for your suggestion. The two plots were added in the main manuscript as Appendix.

The manuscript is generally well-written but includes minor typographical errors (e.g., missing articles or conjunctions). A thorough proofreading is recommended.

A: Thank you for your comment. The English language in the manuscript was polished.

Ethical considerations are adequately addressed. However, authors with disclosed conflicts of interest should clarify how these did not influence the study outcomes.

A: Thank you for pointing this out. Given the aim of study on developing a predictive model for an objective outcome, independent of any device-related factors, the reported disclosures did not impact the study results in any way.

Reviewer 2 Report

Comments and Suggestions for Authors

Congratulation on great quality research. The introduction is clear, the methods provide detailed description of the whole research, results are complete and discussion provides further details on each clinically important factors. The scale proposed in this study should be implemented in everyday practice as useful tool for prediction of postoperative atrial fibrillation.

Author Response

Congratulation on great quality research. The introduction is clear, the methods provide detailed description of the whole research, results are complete and discussion provides further details on each clinically important factors. The scale proposed in this study should be implemented in everyday practice as useful tool for prediction of postoperative atrial fibrillation.

A: We thank the reviewer for his positive words and time to review our manuscript.

Reviewer 3 Report

Comments and Suggestions for Authors

Summary

This manuscript presents the POLARIS score, a predictive model for postoperative atrial fibrillation in patients undergoing cardiac surgery. The authors used a cohort of 3467 patients to construct the model and validated it in an external cohort of 2272 patients. The model stratifies patients into four risk categories and demonstrates moderate predictive power. The study emphasizes the clinical utility of the POLARIS score in identifying high-risk patients for tailored prophylaxis and intervention.

Strengths

POAF is a significant complication in cardiac surgery, and the POLARIS score addresses a critical need for risk stratification.

The use of a large, multicenter dataset enhances the model's generalizability.

The score's simplicity and bedside applicability are key strengths for real-world clinical use.

External validation in a separate cohort strengthens the reliability of the findings.

Recommendations

The model's AUC is low (~ 0.6), raising concerns about its clinical utility. Can the authors explore methods to improve the predictive performance, such as incorporating echocardiographic or intraoperative variables?

The POLARIS score overlaps with existing models in its reliance on well-known predictors (e.g., age, valve surgery). How does this score uniquely improve upon previously published models, such as the CHA2DS2-VASc score?

The manuscript briefly mentions calibration metrics (e.g., Hosmer-Lemeshow test). Would graphical calibration plots help demonstrate the model's reliability across risk categories?

Why is mitral valve surgery the strongest predictor? Could the authors provide mechanistic insights into this association? Why does obesity have a weaker predictive value despite its known contribution to atrial remodeling?

Significant differences exist between the derivation and validation cohorts (e.g., valve surgery rates, baseline characteristics). How do these variations affect the generalizability of the model?

The discussion lacks actionable recommendations for implementing the POLARIS score in clinical practice. How should clinicians use the score to guide prophylactic strategies, such as beta-blockers or amiodarone?

Although the model is externally validated, prospective testing is essential to confirm its utility. Are there ongoing or planned studies to evaluate its real-world application?

Conclusion

The paper offers cardiac surgeons a practical tool by presenting a clinically relevant and well-validated model for predicting POAF. However, its influence is diminished by low-quality numbers, a lack of mechanistic understanding, and limitations in prediction power.

Author Response

This manuscript presents the POLARIS score, a predictive model for postoperative atrial fibrillation in patients undergoing cardiac surgery. The authors used a cohort of 3467 patients to construct the model and validated it in an external cohort of 2272 patients. The model stratifies patients into four risk categories and demonstrates moderate predictive power. The study emphasizes the clinical utility of the POLARIS score in identifying high-risk patients for tailored prophylaxis and intervention.

Strengths

POAF is a significant complication in cardiac surgery, and the POLARIS score addresses a critical need for risk stratification.

The use of a large, multicenter dataset enhances the model's generalizability.

The score's simplicity and bedside applicability are key strengths for real-world clinical use.

External validation in a separate cohort strengthens the reliability of the findings.

A: We would like to thank the reviewer for the compliments and suggestions to improve the manuscript.

Recommendations

The model's AUC is low (~ 0.6), raising concerns about its clinical utility. Can the authors explore methods to improve the predictive performance, such as incorporating echocardiographic or intraoperative variables?

A: Thank you for your conscientious comment. Our AUC of 0.655 would be defined as moderate, not low. We do agree that is not excellent (>0.8), but it is consistent with other described clinical scoring systems described in the Discussion section, which had their own drawback as previously described. Besides, the CHA2DS2-VASc score has been shown in a meta-analysis to achieve AUC values in the range from 0.59–0.71, REF. 33. This demonstrates that an AUC of ~0.65, while modest, aligns with the performance of other tools in literature, but it applies to a population of general cardiac surgery all-comers.

Regarding echocardiographic data, it is important to note that ejection fraction was tested in the univariable regression analysis but was found to be non-significant. Valve insufficiencies were not included in the model because they were closely related to the type of procedure and including them would have introduced multicollinearity. More AF-specific echocardiographic data would be necessary.

As highlighted in the Discussion section, the use of intraoperative variables would hamper the possibility to adopt preoperative POAF prophylaxis.

The POLARIS score overlaps with existing models in its reliance on well-known predictors (e.g., age, valve surgery). How does this score uniquely improve upon previously published models, such as the CHA2DS2-VASc score?

A: Thank you for this important question. The CHA2DS2-VASc score has been shown in a meta-analysis to achieve AUC values in the range from 0.59–0.71, REF. 33. This demonstrates that an AUC of ~0.65, while modest, aligns with the performance of other tools in literature, but it applies to a population of general cardiac surgery all-comers. Moreover, the CHA2DS2-VASc score was originally built for stroke prediction, not POAF.

The manuscript briefly mentions calibration metrics (e.g., Hosmer-Lemeshow test). Would graphical calibration plots help demonstrate the model's reliability across risk categories?

A: Thank you for your suggestion. We do agree with the reviewer and a brief description of Hosmer-Lemeshow test was added in the Discussion and the already present Supplementary Figures of calibration plots were moved in the main manuscript as Appendix.

Why is mitral valve surgery the strongest predictor? Could the authors provide mechanistic insights into this association? Why does obesity have a weaker predictive value despite its known contribution to atrial remodeling?

A: Thank you for pointing this out. The answer to this question is extremely difficult and with our data we cannot answer with certainty, but only provide hypotheses. As already described in the Discussion during mitral valve surgery the left atrium is opened and injury/inflammation to this structure may be the most influential cause of POAF. Obesity, which remains a significant predictor in our model, may possibly have a reduced influence in this study due to the particular type of patients we are dealing with (cardiac surgery patients) and not general population.

Significant differences exist between the derivation and validation cohorts (e.g., valve surgery rates, baseline characteristics). How do these variations affect the generalizability of the model?

A: Thank you for your professional comment. The external validation cohort was intentionally selected to represent a different population, which allows us to assess the generalizability of the model across varied settings. Discrepancies between the cohorts, such as differences in baseline characteristics, were not artificially adjusted, as they reflect real-world variability. This is a strength of external validation, as it provides insight into how well the model performs in populations distinct from the derivation cohort.

The discussion lacks actionable recommendations for implementing the POLARIS score in clinical practice. How should clinicians use the score to guide prophylactic strategies, such as beta-blockers or amiodarone?

A: Thank you for this interesting question. In brief, we are unable to address the reviewer’s question due to the lack of relevant data, as this aspect has not been investigated in the current study.

As mentioned in the "Future Perspectives" section, current guidelines recommend the use of antiarrhythmic drugs for POAF prophylaxis; however, this recommendation is infrequently applied in practice. The POLARIS score aims to help identify higher-risk patients, potentially encouraging physicians to implement such prophylactic measures. However, due to the absence of investigation, the specific therapy for this patient subset remains unknown at this time.

Although the model is externally validated, prospective testing is essential to confirm its utility. Are there ongoing or planned studies to evaluate its real-world application?

A: Thank you for this important question. As mentioned in the final part of the manuscript this study would set the base, after peer-review approval, for future studies to evaluate its real-world application.

Conclusion

The paper offers cardiac surgeons a practical tool by presenting a clinically relevant and well-validated model for predicting POAF. However, its influence is diminished by low-quality numbers, a lack of mechanistic understanding, and limitations in prediction power.

A: Thank you for pointing this out. We acknowledge and agree that the current study has certain limitations, which have been thoroughly outlined. However, it also has significant strengths, such as the inclusion of all-comer patients rather than focusing solely on specific subgroups like CABG patients.

While it would be premature to definitively assert specific mechanisms, the POLARIS score incorporates well-established predictors of POAF, and we have made efforts to justify the importance of certain parameters by referencing existing hypotheses. POAF remains a complex and not fully understood condition. Many other studies have attempted to develop reliable scores, but as mentioned, none have achieved exceptional results, and our study falls within the established range of acceptable performance.

Round 2

Reviewer 1 Report

Comments and Suggestions for Authors

The manuscript has been adequately revised.

Reviewer 3 Report

Comments and Suggestions for Authors

The addition of calibration plots to the main text, more discussion of the AUC in relation to current models, and explanations of important predictors and cohort differences have all significantly enhanced the work. Although there are still limits, they are clearly stated, and the work adds much to the field. The POLARIS score is a practical and validated tool for risk stratification in cardiac surgery patients.